# An Evaluation of the Development Performance of Small County Towns and Its Influencing Factors: A Case Study of Small Towns in Jiangyin City in the Yangtze River Delta, China

**Xiao Gong** [1] , **Xiaolin Zhang** [1,2,*], **Jieyi Tao** [1], **Hongbo Li** [1,2] and **Yunrui Zhang** [3]

1 School of Geographical Sciences, Nanjing Normal University, Nanjing 210046, China; 201302028@njnu.edu.cn (X.G.); 201302046@njnu.edu.cn (J.T.); lihb@njnu.edu.cn (H.L.)
2 Jiangsu Center for Collaborative Innovation in Geographical Information Resource Development and Application, Nanjing 210046, China
3 School of Architecture and Urban Planning, Nanjing University, Nanjing 210093, China; mg20360057@smail.nju.edu.cn
* Correspondence: zhangXiaolin@njnu.edu.cn

**Abstract:** Research on the development performance of small towns is critical for promoting their revitalization, advancing urbanization, and high-quality development and transformation for realizing urban–rural integration. We used the DPSIR-DEA model to study the spatiotemporal evolution process and characteristics of the development performance of 14 small towns within the administrative division of Jiangyin city from 2001 to 2019. We subsequently applied a geographical detector model to analyze the spatiotemporal heterogeneity of the factors influencing the development performance of small towns. The results showed that 2012 was a turning point in the overall development performance index of small towns in Jiangyin, revealing initially decreasing and then increasing trends. The development performance index values of different types of small towns evidenced three trends: a steady increase, a continuous decrease, and an initial decrease followed by an increase. During 2001–2019, the development performance of Jiangyin's small towns reflected a spatial evolution pattern of complete dispersion → small agglomeration → large agglomeration. An optimal spatial pattern comprised an increase in the number of towns demonstrating a high development performance and a decrease in the number of towns with a low development performance. GDP per capita, industrial investments, and construction land density were key influencing factors of development performance, which was mainly driven by economic and social factors, with ecological factors having a relatively weak influence.

**Keywords:** small towns; development performance evaluation; spatiotemporal evolution; influencing factors; Jiangyin City



## 1. Introduction

Small towns, which serve as key links between cities and rural areas, play a unique role in the integration of urban and rural areas and the promotion of China's new-type urbanization and rural revitalization initiatives. Moreover, their construction and development directly reflect the overall political, economic, and cultural character of a region [1]. The unique status of small towns and the role they play in urbanization have been one of the main focuses of scholarly attention for a long time. From the traditional research perspective, it is generally believed that small towns lack economic efficiency with their stagnated social-economic development [2–4]. Therefore, it is also questioned whether the big cities, medium cities, or small towns would turn out to be the main driving force of urbanization in the new era, triggering a dispute worldwide, especially in China [5,6]. In the face of such a debate, small town evaluations have been conducted with the aim of exploring the importance of small towns in the urbanization process. The role played by small

towns in urbanization in various regions and countries around the world is summarized, and these development models and experiences can also be used as a reference for small towns and urbanization in China. For instance, in Japan and South Korea, the national government has implemented industrial revitalization and construction strategies for rural areas at different times, along with the rise of mega cities and the decline of peripheral fringe areas. The thriving development of small fringe town zones has been developed as a result [7]. In Germany, small- and medium-sized towns represent the main population concentrations and are the mainstay of urbanization. The country focuses on the coordination mechanism of balanced development, with few differences in size between towns and cities and thus the achievement of diversified development [8]. In the evolution of urban society in North America, urban space tends to be territorialized, and urban–rural boundaries are gradually broken. Small town areas with competitive new economic characteristics have attracted a large amount of human capital for settlement due to their good community environment [9], driving local urbanization. In addition, a number of individual-specific evaluation studies have also shown significant value. From functional and network systems of small- and medium-sized towns, an environment could be provided wherein people could feel friendlier and less hectic, and with lower housing costs or with more outdoor activities they would have employment opportunities in different locations compared with cities [10]. Furthermore, from the example of urbanization in Germany Bavaria, and rural areas of South Korea that realized their development through rural in situ urbanization (RISU), some researchers advocate that small towns are the future of urbanization [11]. Therefore, regardless of the research perspective, small towns can bridge the gap between urban and rural areas to effectively promote the quality of urbanization [12].

China's urbanization has been a momentous event that has attracted wide international attention [13]. In March 2014, the Central Committee of the Communist Party of China (CPC) and the State Council jointly released a "National New-type Urbanization Plan (2014–2020)" [14]. This was the first official plan to regard new-type urbanization as a national policy [15], which recommended an urbanization approach with Chinese characteristics for achieving the coordinated development of large, medium, and small cities and of small towns, and for comprehensively improving the quality of urbanization. Small towns have become a hot topic in research focusing on China's new-type urbanization strategy. The evaluation of their development performance is important for assessing the quality of urbanization, while also serving as a useful benchmark for improving the quality of regional urbanization.

The study of performance evaluations has always been a critical concern within academic circles in China and abroad. Performance evaluations entail a consideration of the original performance goals as the study criteria and the application of uniform evaluation standards to ensure an objective, fair, and comprehensive evaluation of the outputs of an organization or project within a certain period of time [16–18]. With the gradual conceptual advancement of performance evaluation systems [19–21], scholars at home and abroad have gradually extended their research fields to cover complex systems, such as towns or cities. They have conducted various studies entailing performance evaluations of urban development. Differing from the performance evaluation of a single project or organization, the evaluation of urban development performance focuses on aspects of the economic, social, spatial, demographic, and environmental efficiency of cities or towns during a specific period of time, providing a concrete way of testing the quality of urban development.

Research on urban performance evaluations outside of China has an older history and covers a wide range of fields, mainly from the perspectives of urban social welfare [22], the industrial structure [23,24], infrastructure allocation efficiency [25,26], and policy and institutional management [27,28]. These studies have focused on topics such as the locations of cities and towns [29,30], spatial structures and scale [31,32], key sectors and industrial clusters [33–36], and ecological performance and sustainable development [37–40]. Research methods have entailed a combination of econometric models, spatial models,

and semi-structured interviews [41–44]. In China, studies to assess urban performances have mostly focused on the entire country [45], economic zones [46,47], provinces [48,49], urban agglomerations [50,51], and other spatial levels. Some studies have evaluated operational efficiency at specific levels, including green development efficiency, urban land use efficiency, urbanization efficiency, economic development performance, and industrial efficiency. Others have been aimed at perfecting and innovating the index system used to evaluate urban development performance to improve its scientific basis [52,53]. The research methods used are quantitative as well as qualitative, with qualitative studies mainly centering on discussions of problems and factors influencing urbanization efficiency and policy recommendations to improve urban development efficiency [54,55]. Quantitative studies have mainly centered on data envelopment analysis [45,50] and the use of the comprehensive index method, the gray correlation projection method, and other tools for measuring development performance [56].

In summary, current research on development performance evaluations evidences the following characteristics. First, existing studies on performance evaluations have paid more attention to the study of spatial units at medium and large scales, such as urban agglomerations and the entire country, while neglecting the study of individual small towns. Second, up to now, performance evaluations have mostly targeted a specific aspect of urban development, which can only reflect the performance of specific areas, and have therefore not assessed the overall level of development.

The development performance of small towns is evaluated as an independent composite system of inputs and outputs [57]. It refers to the ratio of the effective outputs of all factors to the overall inputs across diverse economic, social, and ecological fields in the development and construction of small towns under certain conditions of production technology per unit of time. It reflects the effective allocation, rational use, and management of the input resources of small towns in a comprehensive manner [58], concentrating on quality improvements in the development of small towns.

Different views and controversies relating to an understanding of the role and status of small towns have been evident in the implementation of strategies for their development [59,60]. In the actual process of their development, there are also longstanding problems relating to sloppy land use, scattered capital investments, and low development efficiency. The evaluation of small towns' development performance is aimed at improving the quality of their development and optimizing the allocation of resources by using specific technical methods to develop a system of indicators that reflects development performance and by following appropriate procedures to ensure scientific judgments and assessments [61]. Accordingly, targeted, specialized, and featured policy recommendations on the high-quality development of small towns can be made on the basis of the evaluation, which is therefore an important aid for decision making, contributing to improving the level of regional sustainable development and achieving the overall and coordinated development of urban and rural areas.

Southern Jiangsu is one of the core areas of the Yangtze River Delta urban agglomeration. In the 1980s, counties and townships in this area played a leading role in the creation of the nationally acclaimed "Southern Jiangsu Model" through the development of township-based industries and a collective economy and through active participation in market regulation [62]. As one of the birthplaces of the "Southern Jiangsu Model," Jiangyin City has promoted advanced social and economic conditions in small towns and has long evidenced a trend of high-speed development, thus occupying a prominent position in China. A case study conducted in Jiangyin would therefore yield valuable inputs.

Accordingly, in order to understand the impact of the high-quality development of small towns on China's new-type urbanization, we explain the importance of small towns by evaluating their development performance and make relevant policy recommendations based on the results. In this paper, 14 small towns within the administrative division of Jiangyin city, which is located in southern Jiangsu Province, including 9 organic towns, 2 economic and technological development zones, and 3 streets, were selected as the study

site. Using the DPSIR-DEA model, we selected various economic, social, and ecological indicators to perform a comprehensive evaluation of the development performance of small towns and to analyze their spatiotemporal evolution. The geographical detector model was also used to identify factors influencing changes in development performance to strengthen the results of the evaluation of small towns and to provide conceptual as well as policy support for the high-quality development of small towns along with guiding inputs on further studies on the development performance of small towns.

## 2. Materials and Methods

### 2.1. Study Area

Jiangyin is a riverside port city located in the middle and lower reaches of the Yangtze River in southeastern Jiangsu Province. The city, which is equidistant (150 km) from the two major cities of Nanjing and Shanghai, has an administrative area of 987.5 km$^2$. Therefore, its location is highly strategic. Jiangyin has a developed economy and has been consistently ranked among the top 2 of China's 100 counties. In 2019, Jiangyin had a registered population of 1,264,100 and a permanent population of 1,653,400, while the regional GDP was 400.112 billion yuan.

A total of 10 organic towns and 7 streets falls under the city's administration. In addition, the Jiangyin High-Tech Development Zone and Jiangyin Harbor Economic Development District, which are township-level administrative units, are under its jurisdiction. The former includes Chengdong Street, while the latter includes streets and organic towns, such as Lingang Street, Shengang Street, Xiagang Street, and the town of Huangtu. For this study, the small towns examined are all defined as township-level administrative units within the jurisdiction of Jiangyin, covering organic towns and streets and economic and technological development zones established through the merger of one or more organic towns. We focused on the economic and technological development zone as a unified research unit in which internal streets and organic towns were merged as the research criteria.

As a result, the research area thus covers two economic and technological development zones, three streets, and nine organic towns within the administrative division of Jiangyin. These areas are: the Jiangyin Harbor Economic Development District (hereinafter referred to as Harbor Development District) and the Jiangyin High-Tech Development Zone (hereinafter referred to as the High-Tech Zone), the streets of Chengjiang, Nanzha, and Yunting, and the towns of Yuecheng, Qingyang, XuXiake, Huashi, Zhouzhuang, Xinqiao, Changjing, Gushan, and Zhutang (Figure 1).

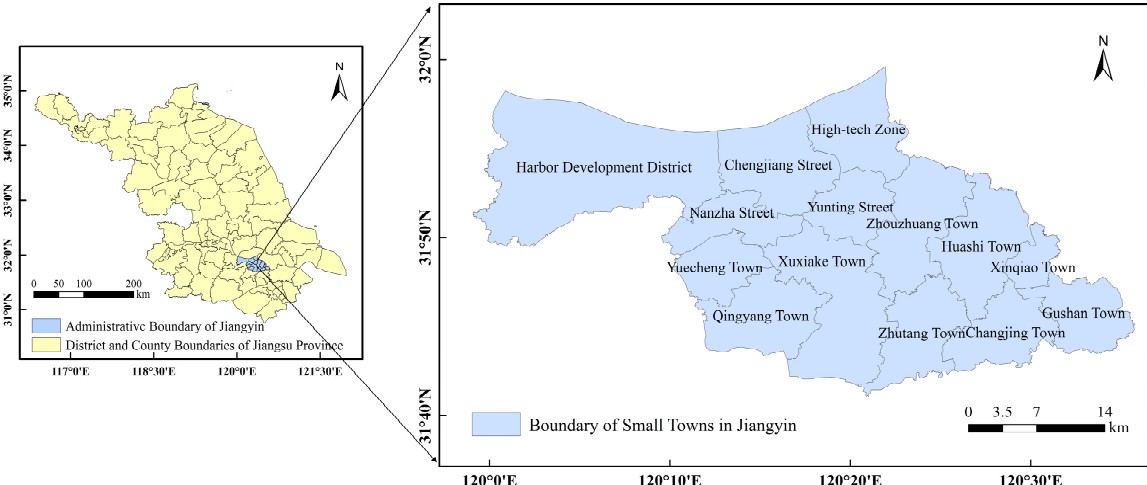

**Figure 1.** Location of areas covered in Jiangyin City.

### 2.2. Data Source and Processing

The study data included statistical panel data, remote sensing image data, digital elevation model (DEM) data, and normalized difference vegetation index (NDVI) data, as shown in Table 1. Population, industrial, and related economic and social data were obtained from the Jiangyin Statistical Yearbook and the Jiangyin Yearbook for the years 2001–2019. ArcGIS zoning statistics, spatial interpolation, and raster calculations were used to obtain data on the NDVI index, the green field rate, elevation, river density, and construction land density. Spatial data on changes in small towns relating to the merger and adjustment of administrative divisions were sourced from the latest small town zoning map and dropped onto the corresponding towns. Consequently, a unified base map of the administrative divisions was obtained.

**Table 1.** Data sources for development performance evaluation indicators in Jiangyin [63–68].

| The Data Name | Year | Data Description | Data Sources |
|---|---|---|---|
| DEM Data | 2019 | Digital elevation model with 30 m spatial resolution | https://www.resdc.cn/data.aspx?DATAID=217/ (last accessed on 8 July 2022) |
| Land-use Data | 2001 2014 2019 | Interpretation of remote sensing monitoring data at a 30 m spatial resolution | https://www.resdc.cn/Datalist1.aspx?FieldTyepID=1,3/ (last accessed on 8 July 2022) |
| River and Lake Datasets | 2019 | River and lake vector datasets | https://www.resdc.cn/Datalist1.aspx?FieldTyepID=1,3/ (last accessed on 8 July 2022) |
| Administrative Division Data | 2019 | Used to extract the study base map | https://www.resdc.cn/data.aspx?DATAID=203/ (last accessed on 8 July 2022) |
| NDVI Data | 2001–2019 | Maximum annual NDVI data at a 30 m spatial resolution | http://www.nesdc.org.cn/sdo/detail?id=60f68d757e28174f0e7d8d49/ (last accessed on 8 July 2022) |
| Statistical Yearbook Data | 2001–2019 | Demographic, industrial, social, and other statistics | Jiangyin Statistical Yearbook |

### 2.3. Methods

#### 2.3.1. DPSIR-DEA Model

The Driving Forces–Pressure–State–Influence–Response (DPSIR) Model delineates system indicators into five components: driving force, pressure, state, influence, and response. This structured theoretical model is widely used in the evaluation of environmental system indicators [69–71]. It has the advantages of comprehensive content coverage and a strong logic, which can fully reflect the two-way relationship between the system and human activities. Consequently, it provides a scientific theoretical basis for the study and measurement of the elements and attributes of complex systems.

Data Envelopment Analysis (DEA) [72] is used to evaluate the work performance of organizations of the same type and is appropriate for use in the performance evaluation of independent complex systems, such as small towns. Specifically, it enables the relative effectiveness and efficiency of decision-making units (DMUs) with multiple input and output elements to be assessed [73]. An optimal endogenous method is applied to determine the weights of each input factor, avoiding subjective factors that may affect the input–output relationship [74].

The combined DPSIR-DEA model covers all of the relevant indicators, enabling a comprehensive analysis of the interactions between social and cultural factors, economic development, and the natural ecology. Moreover, it provides for an objective and accurate calculation of the development performance of small towns, which facilitates comparisons between different regions and compensates for shortcomings of inefficient regions according to local conditions.

To ensure the scientific quality and rationality of the index system, we constructed a development performance evaluation system for small towns using the DPSIR model. The constant returns to scale (CRS) model within the DEA model was used to measure small towns' development performance. Taking each small town as a DMU, we assumed that there were $K$ towns, each with $M$ input indicators and $N$ output indicators. With $x_{km}$ representing the input of the $m^{th}$ resource of the $k^{th}$ town, and $y_{kn}$ representing the output of the $n^{th}$ resource of the $k^{th}$ town ($k = 1, 2, \ldots , K; m = 1, 2, \ldots , M; n = 1, 2, \ldots , N$), the $i^{th}$ town was represented in the CRS-based model as follows [75]:

$$\begin{cases} \min\left[\theta - \varepsilon\left(\bar{e}^T s^- + e^T s^+\right)\right] \\ s.t. \sum_{k=1}^{K} x_{km} \lambda_k + s^- = \theta x_m^i; \sum_{k=1}^{K} y_{kn} \lambda_k + s^+ = y_n^i \\ 0 \leq \theta \leq 1, \lambda_k, s^-, s^+ \geq 0, k = 1, \ldots, K \end{cases} \quad (1)$$

In formula (1): $\theta$ denotes the development performance index of small towns, $\lambda_k$ is a weight variable, $s^-$ is a relaxation variable, $s^+$ is the remaining variable, $\varepsilon$ is a non-Archimedean infinitesimal, and $\bar{e}^T = (1, 1, \ldots , 1) \in E^M$ and $e^T = (1, 1, \ldots , 1) \in E^N$ are unit vector spaces. A larger value of $\theta$ corresponded to a higher performance level. When $\theta = 1$, this means that the production frontier of a small town is optimal, and its outputs have reached an optimal overall efficiency level relative to its inputs.

### 2.3.2. Geographical Detector Model

The geographical detector model is a method of statistical analysis used to identify geospatial heterogeneity and reveal the effects of the underlying driving forces [76]. It is an effective method for detecting spatially distributed consistency and causality between two independent interacting variables [77]. The main manifestation is that if the intensity of a factor has a significant consistency or similarity in spatial distribution with the development performance index, it can indicate that this characteristic factor has a significant influence on the development performance index. The degree to which the probe factor $X$ explains the spatial differentiation of $Y$ can be measured by the q-value, which is expressed as follows [78]:

$$q_{X,Y} = 1 - \frac{1}{n\sigma^2_Y} \sum_{j=1}^{m} n_{X,i} \sigma^2_{Y_{X,i}} \quad (2)$$

In formula (2): $Y$ denotes the development performance index of a small town, $q_{X,Y}$ is an explanatory power indicator for the development performance index influencing factor $X$, $n$ is the number of small towns in the study area, $m$ is the number of types of influencing factors, $n_{X,i}$ is the number of small towns within type $i$ for the influencing factor $X$, $\sigma^2_Y$ is the variance in the development performance index of small towns in the study area, and $\sigma^2_{Y_{X,i}}$ is the variance in the development performance index of small towns in type $i$. The value of $q_{X,Y}$ ranges from 0 to 1, and a larger value of $q_{X,Y}$ corresponds to the stronger explanatory power of the $X$ factor regarding the spatial distribution of small towns' development performance.

## 3. Results and Analysis

### 3.1. Evaluation of the Development Performance of Small Towns

#### 3.1.1. Construction of the Index System and Weight Analysis

We drew on previous findings derived from the construction of a quantifiable, comparable, and accessible index system [57] in combination with an assessment of the actual situation in the study area. As Table 2 shows, the DPSIR model was used to select 13 indicators from the input and output levels to construct an index system for evaluating the development performance of small towns in Jiangyin. All indicators can be divided into positive and negative indicators according to their attributes. The symbol of "+" in Table 2 represents positive indicators, which means the higher the value, the greater the weight given to the indicator, and the symbol of "−" is just the opposite. The selected indicators covered the economic, social, and natural ecological characteristics of small towns. Input

indicators included capital and labor factors and resource elements, constituting the driving force, pressure, and state system layers. Output indicators included the economic scale, income level, and ecological benefits, constituting the influence and response system layers.

**Table 2.** Index system for evaluating the development performance of small towns in Jiangyin and indicator weights.

| System Layer | Subsystem | Indicators | Properties | Weight |
|---|---|---|---|---|
| Driving Force(D) | Economic development | Total investment in fixed assets ($10^8$ yuan) | Input indicator (+) | 0.10 |
| | Social development | Year-end employed population ($10^4$ person) | Input indicator (+) | 0.08 |
| | | Population growth rate (%) | Input indicator (+) | 0.14 |
| Pressure(P) | Resource stress | Year-end arable land (acre) | Input indicator (+) | 0.02 |
| | | Industrial electricity consumption ($10^8$ kWh) | Input indicator (−) | 0.06 |
| Status(S) | Investment and construction | Proportion of employees in the secondary industry (%) | Input indicator (−) | 0.04 |
| | | Proportion of employees in the tertiary industry (%) | Input indicator (+) | 0.05 |
| Influence (I) | Economic quality | GDP growth rate (%) | Output indicator (+) | 0.12 |
| | Life quality | Per capita disposable income (yuan). | Output indicator (+) | 0.12 |
| | Industrial structure | The proportion of the secondary industry (%) | Output indicator (−) | 0.10 |
| Response(R) | Consumption mode | Per capita fixed asset stock (yuan) | Output indicator (+) | 0.08 |
| | Ecological resource | Comprehensive energy consumption of industrial enterprises above designated size (tons of standard coal/$10^4$ yuan) | Output indicator (−) | 0.07 |
| | | NDVI index | Output indicator (+) | 0.02 |

Indicators were selected according to five systems. The first was a driving force system comprising two subsystems: economic development dynamics and social development dynamics. In this system, total investments in fixed assets reflected the speed and scale of fixed asset development, representing the quality of industrial development in small towns. The year-end employed population and population growth rate reflected the social attractiveness and quality of urbanization [79]. These indicators constitute potential triggers for changes in the development performance of small towns, which, in turn, collectively exert pressure on the system.

The second system was a pressure system composed of a resource pressure subsystem, which represented the pressure on small towns to achieve a green economy and sustainable development, impacting changes in their development status. Year-end arable land reflects the contradiction between land supply and demand and shows the degree of coordination between local construction and the development and protection of arable land resources. Industrial electricity consumption is one of the important indicators for assessing the degree to which new-type industrialization as well as energy conservation and emission reduction are promoted.

The third system was a state system comprising investment and construction status subsystems, which directly affect the economy, society, and natural resource base of small towns. Notably, the proportion of employees in the secondary and tertiary industries reflects the degree of modernization of industries, resulting from the investment of capital and industrial transformations in small towns.

The fourth system was an influence system comprising subsystems of economic quality and life quality, which prompt small towns to take a series of positive measures to respond to changes in the external environment. The GDP growth rate and per capita disposable income are closely related to the wage income level of urban residents, which is the most intuitive embodiment of the development benefits of small towns. It can reflect the degree of progress of small towns in terms of their economic development and social security during a certain period of time.

The fifth system was a response system comprising the following subsystems of responses: the industrial structure, consumption mode, and ecological resources. The

proportion of the secondary industry is an important indicator reflecting the rationality of the industrial structure [80,81] and is used to determine the types and patterns of small towns. The per capita fixed asset stock indicates the ability of urban residents to cope with economic risks, directly reflecting consumption patterns, the quality of urban residents' lives, and the level of economic development of small towns. The comprehensive energy consumption of industrial enterprises above a designated size directly reflects the degree of dependence of industrial development on energy resources [82]. The NDVI index [83] covers the crop growth status and ecological vegetation cover. These indicators commonly reflect the response of ecological and energy resources to the construction of small towns.

The entropy method was used to assign weights to each indicator. The results showed that from the perspective of the system layer, the driving force weight was 0.32, the pressure weight was 0.08, the state weight was 0.09, the influence weight was 0.24, and the response weight was 0.27. The driving force, influence, and response systems evidently played influential roles in the development performance of small towns in Jiangyin, mainly in terms of economic and social indicators. The weight of the state system layer was close to the weight of the pressure system, and their degrees of action were comparable.

A total of 5 of the 13 indicators, all of which were economic, industrial, and social indicators, with weights above 0.1 and a total weight of 0.58, had the greatest impact on the development performance of small towns in Jiangyin. The driving force and influence systems occupied two indicators separately. The indicators within these two systems have a significant impact on the development performance of small towns in Jiangyin, revealing that positive measures taken in these towns to promote economic and social development have had positive effects on their development performance such as increased investments, greater talent attraction, and improved incomes.

Our results indicated that there were seven input indicators and six output indicators, evidencing a balance in their numbers. The total weight of all of the input indicators was 0.49, while the total weight of all of the output indicators was 0.51, indicating that the research system has maintained a stable input–output structure over time, revealing the existence of a scientific foundation for the evaluation of the development performance of small towns in Jiangyin.

3.1.2. Analysis of Development Performance Trends

We used the DEAP software, version 2.1, to measure the development performance index of 14 small town units in Jiangyin from 2001 to 2019. Figure 2 depicts the comprehensive development performance index of each town, while Figure 3 shows the development performance trend curve for small towns, which more intuitively reflected the changes in their development performance.

From the overall perspective, the average development performance of small towns in Jiangyin shows a uniform, gradual decreasing trend followed by an increasing trend. Because there was relatively little overall fluctuation, the overall development of small towns in Jiangyin was relatively stable during the period 2001–2019. The average value shows a decrease prior to 2012 and an increase after 2012. The main reasons are as follows. On the one hand, at the turn of the 21st century, when the process of regional globalization was deepening, the external environment for small town development was becoming negative, and the dominant and supporting role of township enterprises was beginning to be questioned [84,85]. Consequently, the traditional "Southern Jiangsu model" gradually fell into decline and went downhill, no longer adapting to the needs of economic development at that time. During this period, the "New Southern Jiangsu Model" that ushered in the transformation had not yet been explored, and the new reform of the economic system was not effective, leading to a gradual decrease in the overall development performance of small towns in Jiangyin in the early 21st century. On the other hand, around 2012, with the development system associated with the New Southern Jiangsu Model reaching maturity, the authorities actively seized the developmental opportunity for advancing the reform of the economic system and management of towns and villages, relying on Shanghai's active

leadership role in introducing foreign investments and facilitating traditional township enterprises in upgrading to high-tech industries. During this period, many administrative changes led to the growth of townships, the gradual improvement of industrial parks, and the increasing maturity of featured industries in small towns. At the same time, large quantities of land for construction were available, and economic and social development was reaching a new normal level, revealing the urbanization characterized by capital and manufacturing, along with the construction of development zones being the main driving force. These are the explanations for the trends in the average development performance of small towns.

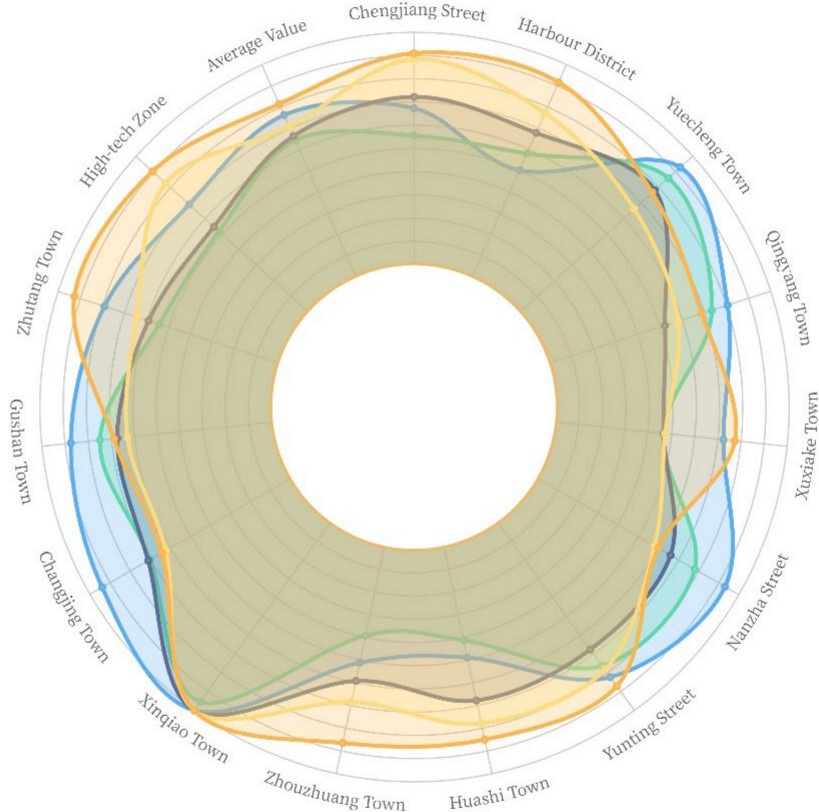

**Figure 2.** Development performance index of small towns in Jiangyin.

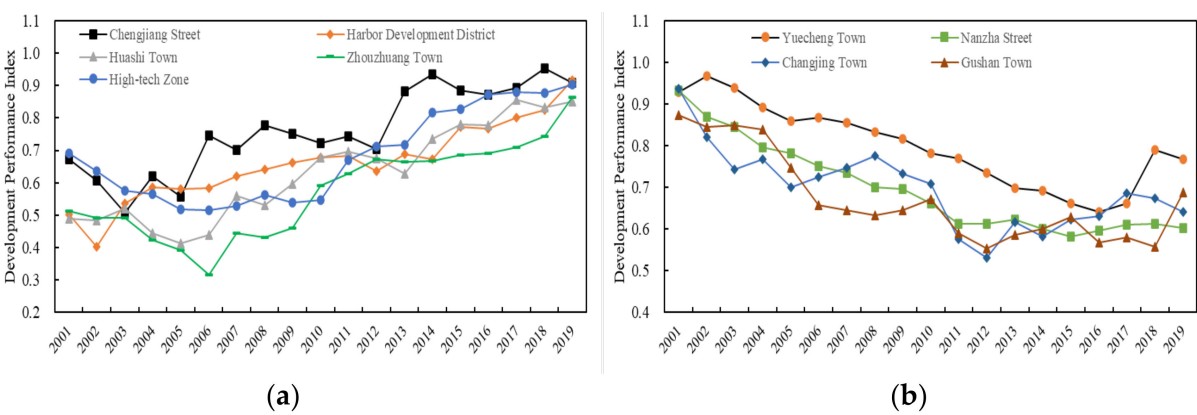

**Figure 3.** *Cont.*

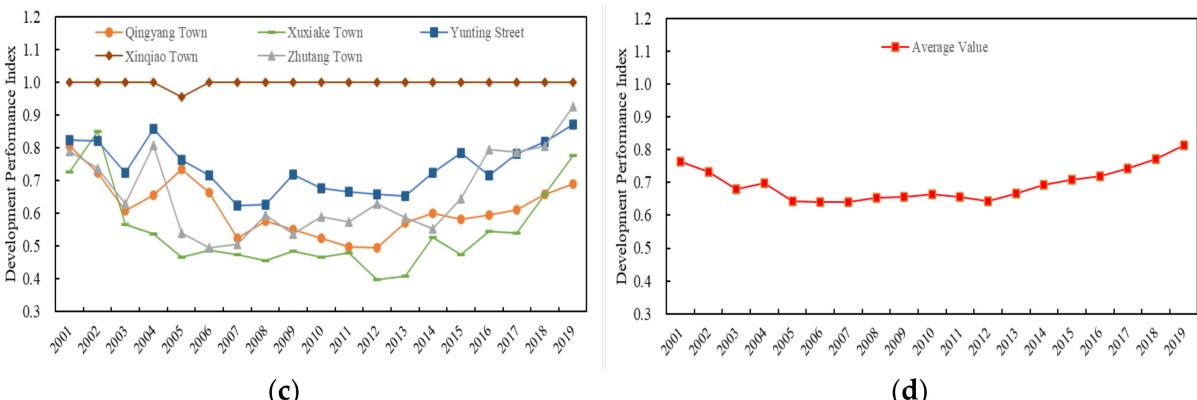

**Figure 3.** Trends in the development performance of small towns in Jiangyin from 2001 to 2019. (**a**) Overall rise (**b**) Overall decline; (**c**) Initial decrease and then increase (**d**) Trend of average values.

From the perspective of individuals, three trends can be discerned in the development performance of small towns in Jiangyin from 2001 to 2019. The first is a stable overall increasing development performance trend for Chengjiang Street, Harbor Development District, the High-Tech Zone, and the towns of Huashi and Zhouzhuang. Of these areas, Chengjiang Street, Harbor Development District, and the High-Tech Zone are located in or near the central city of Jiangyin, whereas the towns of Huashi and Zhouzhuang are industrial towns located in the eastern part of Jiangyin City. These small towns have conducive conditions for the economy and society, and are strong industrial towns with well-established industrial and service industries, which epitomizes the Southern Jiangsu Model. They merged with neighboring towns during the many administrative changes, and have acquired more resources. Consequently, they are strongly cohesive and expansive in terms of their scale and economic strength, and their development performance is steadily increasing.

The second is an overall decreasing development performance trend for Nanzha Street and the towns of Yuecheng, Changjing, and Gushan. These small towns all have strong eco-agricultural characteristics and beautiful scenery, and are relatively far away from the city center. Moreover, primary industries are more evolutionary in these areas. They mostly developed traditional secondary industries such as the chemical industry and manufacturing, which had certain advantages during the initial process of transitioning into a planned economy. However, they were also limited by their scale and policy support. Consequently, they had not cultivated a good self-generating mechanism and atmosphere for enterprises, resulting in the absence of a long-term driving force to achieve industrial transformation and attract talent and investments. With the gradual improvement of the market economy and intensification of competition, these towns were left behind in the second venture of the township enterprises, resulting in the continuous decrease in their development performance each year.

The third is an initially decreasing and then increasing development performance trend for Yunting Street and the towns of Qingyang, XuXiake, Zhutang, and Xinqiao. The development performance of such small towns is generally similar to the average development performance trend of small towns in Jiangyin, taking 2012 or so as a turning point. These small towns can be divided into two types to explain the reasons for the trends in their development performance.

Yunting Street and the towns of Qingyang and Zhutang are traditional industrial towns dominated by the textile and garment industry and electrical materials. During the initial phase of the 21st century, their total economic volume and social development were positioned at the middle and lower ends for small towns in Jiangyin, and urbanization lagged behind that of other towns. These small towns evidenced economic weaknesses, such as part-time agriculture, a lack of support for tertiary industries, and difficulty obtaining employment. Therefore, a decreasing trend in their development performance was

shown. Around 2012, an expansion mechanism for the new industrial zone gradually took shape, generating clusters with competitive advantages, which resulted in an exponential rise in development performance. This shift was, on the one hand, strongly influenced by locational conditions of being close to the city center, in line with the general trend in the regional environment of the transformation and upgrading of industrial structures and reforms of the economic system resulting from the implementation of a favorable policy. On the other hand, it was also driven by the overflow of population, capital, and industrial transfers from developed towns such as Chengjiang Street and the High-Tech Zone.

Cultural tourism is a feature of the town of XuXiake, whose tourism industry is planned, constructed, and integrated with the famous ancient Chinese geographer Xiake Xu as the core figure, with particular cultural attributes. In the early stage of development, the town's tourism attributes were inferior in relation to the surrounding 5A-class scenic spots, which have both historical backgrounds and sightseeing value, notably the ancient town of Zhouzhuang in Suzhou and YuanTouzhu Park in Wuxi. Consequently, the town's ability to attract tourists was relatively weak, leading to a declining development performance. When the "Outline of the Thirteenth Five-Year Plan for the National Economic and Social Development of Jiangyin (2015–2020)" [86] clearly proposed a goal of taking the provincial XuXiake Leisure Tourism Resort as the carrier to achieve the agglomeration and development of tourism projects in the town of XuXiake afterwards, policy support thus turned into development advantages, resulting in the gradual upgrading of rural tourism by promoting the "tourism +" model, such as "tourism + agriculture" or "tourism + manufacturing" and so on, showing an upward trend in its development performance.

It is noteworthy that the town of Xinqiao is the only small town that has maintained its development performance at a value of 1 over a 19-year period. It is the only specialty town in Jiangyin that engages in intensive land use and applies the model of "three concentrations" (land concentration with regards to the scale of the operation, concentration of peasant residences in township areas, and concentration of enterprises in industrial areas). It also let major enterprises play the role of a "bellwether", driving relations of production, cooperation, and competition among medium and micro enterprises. It has ranked first in Jiangyin in terms of its per capita output and per capita profit creation for quite a long time and achieved an efficient balance between resource inputs and outputs.

### 3.1.3. Spatiotemporal Evolution of Development Performance

In the above analysis, it is apparent that 2012 was a turning point in the change trend for the development performance of small towns in Jiangyin. To show the impact of the measures taken after 2012 as the turning point, we chose the forward year 2014 as the intermediate time point. Therefore, we selected data for 2001, 2014, and 2019 when assessing the combined development performance index of small towns in Jiangyin over the period 2001–2019 by using the natural breakpoint method. Five area types of development performance were used: low-performance areas, relatively low-performance areas, general-performance areas, relatively high-performance areas, and high-performance areas. ArcGIS, version 10.2 was used for spatially visualizing these area types (Figure 4) and for conducting a deeper analysis of the spatiotemporal evolution process.

As Figure 4 shows, in 2001, the central and industrial towns of Chengjiang Street, the Harbor Development District, the High-Tech Zone, and the towns of Zhouzhuang, Huashi, and Xuxiake comprised the low-performance and relatively low-performance areas in 2001, with a wide range of townships situated along the east–west and north–south sides of Jiangyin. Despite having a strong industrial base and a large population and economic scale, these small towns were constrained by technical imitations, a backwards economic structure, and a low resource utilization efficiency that caused them to be unable to transform invested capital, energy, and human resources into high-quality economic and social benefits. The general performance areas were mainly located around areas with high-performance index values (including relatively high-performance areas and high-performance areas), which were traditional industrial towns such as Yunting Street

and the towns of Qingyang and Zhutang. These small towns were subject to the industrial undertaking of developed small towns and the outside world, and have certain enterprise bases. However, the land use in the township area was fragmented and unable to capitalize on its development advantages. Areas with high-performance index values were mainly distributed in the southeast and west of Jiangyin, which are far away from the urban area, and are mostly eco-agricultural towns, such as Nanzha Street and towns of Yuecheng, Xinqiao, Changjing, and Gushan. Benefiting from natural environmental resources, these small towns' agricultural industries were more developed, and the continuous development of ecological land, such as farmland and water bodies, resulted in intensive land use in the town area, which was subject to technological changes from an early stage. Higher levels of development performance were evident in areas with less restrictive conditions. In general, the development performance of small towns in Jiangyin in 2001 evidenced a decentralized spatial pattern of "high on both sides and low in the middle".

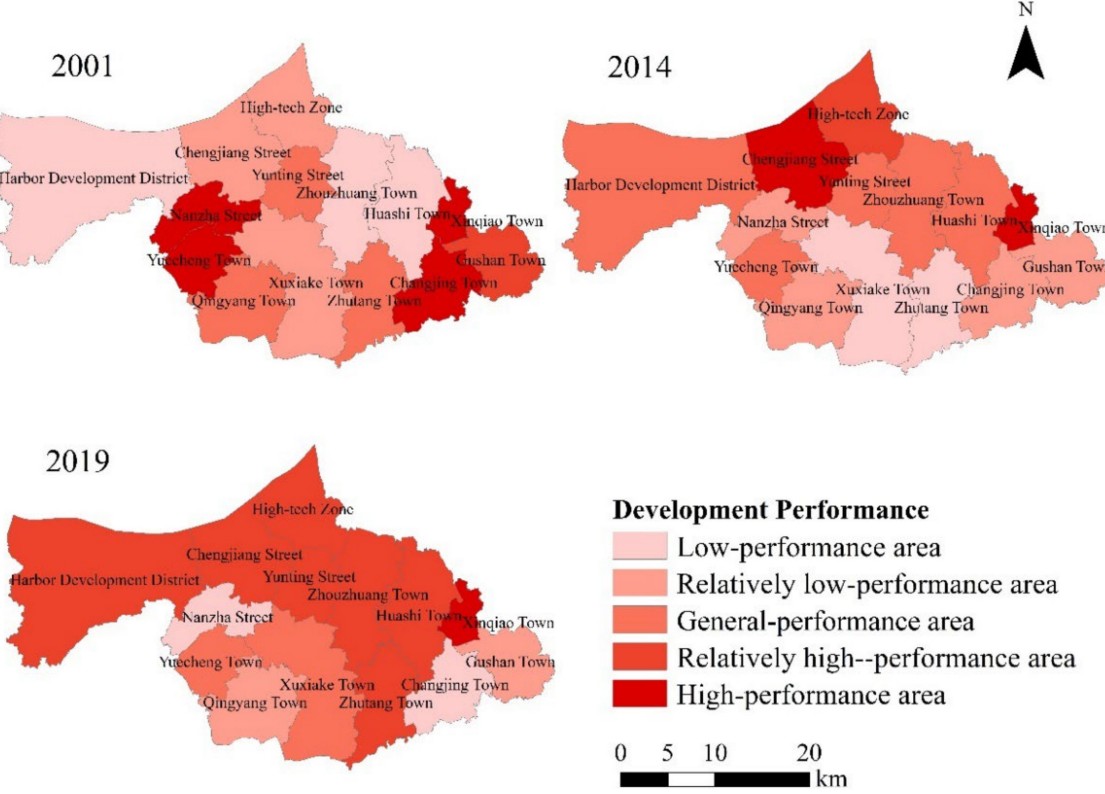

**Figure 4.** Spatial patterns of the development performance of small towns in Jiangyin from 2001 to 2019.

In 2014, after several rounds of technological innovation and capital introduction had been completed in Jiangyin, the development model changed and the spatial pattern of the development performance of small towns underwent tremendous changes, gradually transforming into a spatial pattern of small clusters decreasing from the northeast to the southwest. During this period, the development performance levels of several small towns changed. Areas such as Chengjiang Street, Harbor Development District, the High-Tech Zone, and the towns of Zhouzhuang and Huashi have been upgraded respectively from low-performance and relatively low-performance areas to areas with general and high development performance index values. All of these towns are located in or near the central area of Jiangyin and are intersected by several major transportation routes. Because of their superior locations and transportation conditions, these small towns maintain close ties in terms of resource allocation and flows. Apart from that, with the rapid progress of science and technology, the limitations on industrial development at the technical level have gradually been compensated for, and the transformation and upgrading of traditional

enterprises to high-tech enterprises has effectively improved Jiangyin's development performance. There are also some areas, such as Nanzha Street and the towns of Qingyang, Changjing, and Gushan, which evidenced a decline from being relatively high-performance and high-performance areas to areas with low development performance index values (lower-performance and relatively low-performance areas). Their overall ranking was lower than that of other small towns in Jiangyin, and these towns had natural disadvantages in terms of geographical location and the scale of the townships. They received less support in terms of policies and funds, which led to a gradual widening of the gap with developed towns and the appearance of various development problems.

In 2019, the development performance of small towns in Jiangyin presented the spatial pattern of a large agglomeration that took the north as the pole and was high in the middle and low on both sides. The development performance of small towns ranged from low to high, with the town of Changjing and Nanzha Street as the east and west boundary, separately forming a strong circular agglomeration toward the center. As relatively high- and high-performance areas, Chengjiang Street, Harbor Development District, the High-Tech Zone, the towns of Zhouzhuang, Huashi, and Xinqiao constructed various types of industries in the town parks in proximity to each other, forming a spatial pattern of contiguous development. In fact, the benefits of agglomeration and scale were expanding through a snowball effect during an advantageous cycle. Areas with general performance index values, such as Yunting Street and the towns of XuXiake and Zhutang gradually took off after the transformation that occurred around 2012, and the development pattern at this stage was similar to that of areas such as Chengjiang Street during the previous stage, showing a general positive trend. Small towns in low-performance areas and relatively low-performance areas, such as Nanzha Street and the towns of Changjing, Gushan, and Qingyang were constrained by backward and homogeneous industrial patterns and showed a slow-down in their economic development. The long-term outflow of the population from these small towns led to a lack of impetus to pursue their economic and social development, creating a vicious circle. Moreover, given the size of the township, there is insufficient space for outside industries and an evident fragmentation of the existing industrial land. In the current situation characterized by an unsustainable supply of land and labor and increasing pressure on the environmental carrying capacity, the inertial dependence of traditional development paths has become a huge obstacle to transformation, and the development process of small towns has fallen into a bottleneck.

Overall, from 2001 to 2019, the spatial pattern of the development performance of small towns in Jiangyin evidenced dramatic changes, from a decentralized spatial distribution to an agglomerated spatial distribution. The number of small towns in low-performance and relatively low-performance areas, which were mostly eco-agricultural towns, decreased from 6 in 2001 to 5 in 2019. The number of small towns in high-performance and relatively high-performance areas, most of which were central and strong industrial towns as well as traditional industrial towns that had benefited from certain changes, increased from 5 in 2001 to 8 in 2019. Development levels continued to improve, indicating that the development performance of small towns in Jiangyin achieved optimal spatial patterns.

### 3.2. Factors Influencing the Development Performance of Small Towns

3.2.1. Analysis of Factor Detection Results

Imbalances in resource endowments and levels of economic and social development affecting each of the small towns in Jiangyin clearly led to spatial heterogeneity in their development performance. To explore the factors accounting for differences in the development performance of small towns in Jiangyin, we selected a total of nine economic, social, and ecological factors considering the profile of the study area and theories of urban systems and urban–rural relationships, as well as the research practices of various scholars who have worked on urban development performance [50–52] (Table 3).

**Table 3.** Factors influencing the development performance of small towns in Jiangyin.

| Influencing Factors | Detection Factors | Factor Interpretation | Unit |
|---|---|---|---|
| Economic factors | X1 GDP per capita | Gross regional product per capita | $10^4$ Yuan |
| | X2 Fiscal revenue | Net income from fiscal funds for the whole year | $10^8$ Yuan |
| | X3 Industrial investment | Total annual industrial industry capital investment | $10^8$ Yuan |
| Social factors | X4 Total social electricity consumption | Sum of the annual electricity consumption of the whole society | $10^8$ KWH |
| | X5 Population density | Number of people in unit area | Person/$km^2$ |
| | X6 Construction land density | Scale of construction land in unit area | % |
| | X7 Terrain elevation | Average elevation of terrain in the region | m |
| Ecological factors | X8 River density | Length of the river in unit area | m/$km^2$ |
| | X9 Greenfield rate | Ratio of greenfield area to total land area | % |

Using the factor detection tool in the geographical detector model, we analyzed the degree of influence of each factor on the spatial differentiation of development performances in 2001, 2014, and 2019. The factors passed the significance test at the 0.05 level for each year.

It can be seen from Table 4 that in 2001, industrial investments (X3), with a q-value of 0.7381, had the greatest influence on the development performance of small towns in Jiangyin, while population density (X5) had the least influence, with a q-value of only 0.2829. From a systemic perspective, economic factors have the greatest influence on the development performance of small towns. The significant influencing factors for the development performance of small towns in Jiangyin in 2001 were confirmed by the high rankings of the three economic factors (2nd, 4th, and 1st for X1, X2, and X3, respectively). Both social and ecological factors had a certain degree of influence on development performance, but this influence was not as high, and differences in q-values were not so large, indicating that the effects of social and ecological factors on the development performance of small towns in Jiangyin were similar in 2001.

**Table 4.** Detection results of factors influencing the development performance of small towns.

| Influencing Factors | 2001 | | 2014 | | 2019 | |
|---|---|---|---|---|---|---|
| | $q_{X,Y}$ | $q_{X,Y}$ Ranking | $q_{X,Y}$ | $q_{X,Y}$ Ranking | $q_{X,Y}$ | $q_{X,Y}$ Ranking |
| X1 | 0.4768 | 2 | 0.6400 | 2 | 0.7661 | 1 |
| X2 | 0.4208 | 4 | 0.3941 | 6 | 0.3761 | 6 |
| X3 | 0.7381 | 1 | 0.6711 | 1 | 0.5794 | 4 |
| X4 | 0.3757 | 7 | 0.5770 | 4 | 0.6588 | 3 |
| X5 | 0.2829 | 9 | 0.2763 | 8 | 0.2242 | 8 |
| X6 | 0.4482 | 3 | 0.6348 | 3 | 0.6600 | 2 |
| X7 | 0.3890 | 6 | 0.2768 | 7 | 0.2115 | 9 |
| X8 | 0.3929 | 5 | 0.2673 | 9 | 0.2830 | 7 |
| X9 | 0.3252 | 8 | 0.4177 | 5 | 0.5229 | 5 |

In 2014, industrial investment (X3) was still the most influential factor, with a q-value of 0.6711. The factor with the least influence at this time was river density (X8), with a q-value of 0.2673. Among the economic factors, fiscal revenue (X2) showed decreased influence. Among the social factors, the rankings of all influencing factors improved or remained the same. Conversely, the rankings of the other two ecological factors decreased except for the greenfield rate (X9). This result shows that economic factors should not be the only criteria for determining the final outcome of the comprehensive development performance of small towns. Overall, social factors gained in importance, and scientific and technological progress enabled small towns to overcome various problems relating to

natural conditions in the process of development, which led to the consequent decrease in the impact on ecological factors.

In 2019, GDP per capita (X1) was the most influential factor, with a q-value of 0.7661, while terrain elevation (X7) remains the least influential factor, with a q-value of 0.2115. As small towns gradually approached the new normal in the development and transformation process, the rankings of the influence of various economic, social, and ecological factors did not change significantly. This finding indicates that the development performance of small towns was the result of multifactorial influences. Dominance by unilateral factors was impeded, and diversified directions emerged as the main target for the future development of small towns.

### 3.2.2. Analysis of Interactive Detection Results

We further performed interactive detection on all factors and analyzed the degree of influence of various factors on the spatial patterns of the development performance of small towns in Jiangyin. The results of the analysis are shown in Figure 5. The strengths of the interactions between GDP per capita (X1) and construction land density (X6); industrial investment (X3) and construction land density (X6); total social consumption of electricity (X4) and population density (X5); construction land density (X6) and terrain elevation (X7); and construction land density (X6) and greenfield rate (X9) were all at or near 0.9. These results indicate that the interactions between these factors are consistent with the development performance of small towns in Jiangyin. In particular, the interactions between construction land density and other factors were dominant, indicating that the rational development and utilization of construction land in the context of the actual situation of small towns plays an important role in their development performance. A comparison of the interaction detection results at the three time points revealed that the mean values for the interactions of industrial investment (X3), construction land density (X6), and GDP per capita (X1) with other factors were the highest at 0.7924, 0.7673, and 0.7556, respectively. These results indicated that these three factors critically influence the development performance of small towns in Jiangyin.

The interaction values of industrial investment (X3) and GDP per capita (X1) with other factors were the highest for the degree of improved interaction at 0.0938 and 0.0741, respectively. Therefore, compared with the influence of single factors, the interaction between the two factors and other factors had a greater degree of influence on the spatial differentiation of the development performance of small towns in Jiangyin. The explanatory power of the interaction of the two factors was also stronger than that of the single factor, and the type of interaction among the influencing factors was non-linearly enhanced. This explanatory strength gradually stabilized from 2001 to 2019, indicating that the factors did not exert influences independently of each other. Rather, their influence was characterized by synergistic enhancement, indicating that the development performance of small towns in Jiangyin was the result of the nonlinear coupling of multiple factors.

In general, economic and social factors were the main drivers of the development performance of small towns in Jiangyin, and they were also the main factors influencing spatial variations in their development performance. The degree of influence of ecological factors on the development performance of small towns was relatively weak, mainly because of the impacts of the advancement of science and technology, leading to the gradual weakening of constraints associated with the topography and other natural conditions on the development and construction of small towns. However, the role of ecosystems cannot be completely ignored in the development of small towns, and ecological issues evidently require more attention. The results of the analysis revealed that the importance of the green space ratio is gradually becoming recognized and reflected in the development of small towns. In the future, ecological issues associated with the development of small towns will become increasingly prominent as an important factor determining the ability of small towns to achieve sustainable development.

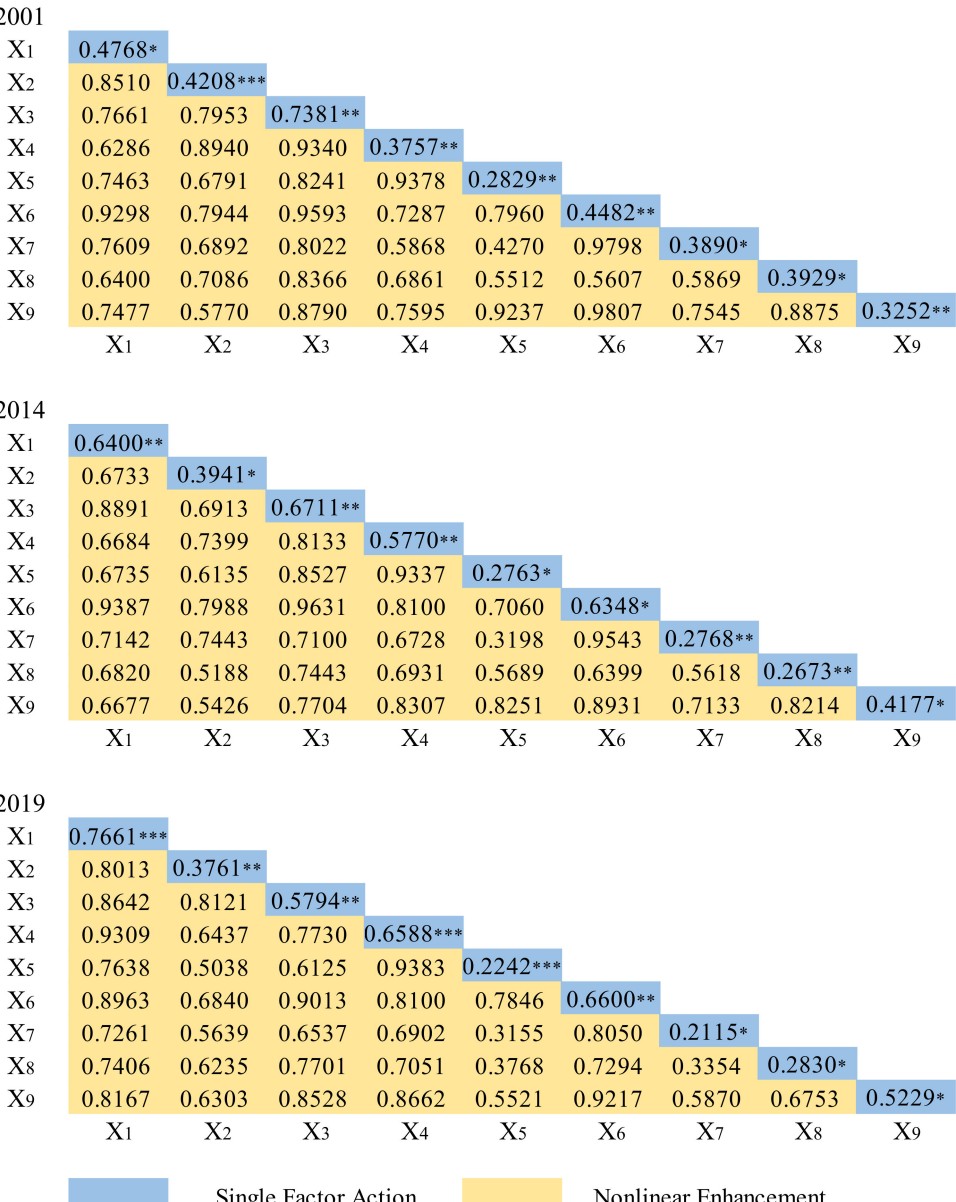

**Figure 5.** Interactive detection results for factors influencing the development performance of small towns in Jiangyin. * represents significance at the 0.05 level, ** represents significance at the 0.01 level, *** represents significance at the 0.001 level.

## 4. Discussion

Taking Jiangyin City as our study area, we analyzed the development performance and spatiotemporal evolution characteristics of small towns in a developed county in the Yangtze River Delta. Moreover, we explored the factors influencing their development performance, thus addressing an existing gap in research on the development performance of small towns at this scale. The findings of this study are of referential and innovative value.

### 4.1. Reliability of Research Results

The DPSIR-DEA model applied in this study combines the advantages of the DPSIR and DEA models and has great scientificity in the construction of the index system and the measurement of development performance index values. From the research results, it is evident that the development performance trends of small towns in Jiangyin and in the coordinated development of industrialization and urbanization in Jiangyin County [87] have been relatively consistent. In both cases, 2012 marked a turning point in the towns'

development, which subsequently showed a trend of continuous improvement. The results for our evaluation of the development performance of the town of Xinqiao were strongly consistent with those of a previous evaluation of the town's sustainable development [88], which confirms the reliability of this study. In addition, there are also related studies that have been conducted on the efficiency of a sample of towns in Jiangsu Province, of which the findings indicated that significant differences do exist in the efficiency characteristics of different types of towns, and the economic development level, input–output efficiency and economic density can be high in small towns [12]. Their research findings are certainly highly consistent with the findings of our study. It also proves that small towns can play an important role in integrating urban and rural development, accelerating new-type urbanization, and promoting a rural vitalization strategy. In terms of the high-quality development of small towns, some studies also concluded that the agglomeration economy, place-based specialization, industrial value creation, and state-led platform urbanism at the small-town-like scale have been positioned at the core of the small town strategy [89,90], which is similar to the analysis of this paper, that small towns need to transform and innovate in industry and policy. Following the policy reform and ongoing innovations in science and technology, small towns in Jiangyin will undergo further transformations in their economic and social development in the future, thus achieving the goal of high-quality development. Moreover, the variability in their development performance associated with spatial distribution will gradually decrease, and their development will shift from being scale-oriented to becoming performance-oriented.

### 4.2. Factors Influencing the Development Performance of Small Towns

This study incorporates economic development zones into the concept of "small towns", thus introducing an innovation into the study of small towns. There are three reasons for this approach. First, economic development zones have been incorporated into the development plans of small towns in most parts of China, and their spatial as well as economic and social development dimensions have long been considered important. Second, because of their wide coverage, economic development zones often contain several contiguous towns, which are highly consistent in terms of their industrial structures and available policy support. Consequently, their overall development levels do not vary greatly. Therefore, as comprehensive representations of the development status of small towns within a continuous territory, economic development zones should be considered as an appropriate research unit. As one of the developed cities in southern Jiangsu, Jiangyin plays an important role as a leader in the development of small towns and shows distinctive characteristics associated with its current economic structure and policy system that are superior for future town planning and development in the national context. Therefore, a third reason for the selection of an economic development zone as a research unit relating to small towns is that it more accurately reflects the actual development of Jiangyin at the local level and has practical implications.

Because of limitations in accessing some data, the indicators for small towns' development performance and influencing factors selected at the ecological level were inadequate. Consequently, the influence of ecological factors on the development performance index values of small towns in Jiangyin was not apparent. Compared with their actual development, there may have been some errors caused by these factors, which need to be augmented in the future. In addition, this study was premised on an objective standpoint and did not take into account the subjective wishes of town residents. In a future study, we will include subjective indicators, such as residents' happiness, by incorporating semi-structured interviews and other research methods within a more in-depth study.

### 4.3. Policy Recommendations

The following recommendations emerged from our findings. First, at the political dimension, the designation of a new-type city in China named a "county-serviced city" (CSC, xian guan shi) [91] can be advocated for qualified small towns within the existing

administrative hierarchy. In the CSC model, small towns would get the same rights as all other large cities in China in dealing with their economic and political development, while maintaining their current position as a township unit in the administrative system and continuing to be served with social service public goods by their county government. Small towns in Jiangyin thus can gain enough autonomy in selling land, planning their future, and managing their development to meet some of the opportunities and challenges they will likely face. We can also learn from the management models of Japan and Germany, where small towns can have autonomy in matters directly related to the daily lives of their residents, including education, welfare, health, finance, etc. [92,93] Moreover, policies directed at the developmental level and bit order of small towns should be formulated scientifically and rationally, and the development target positioning of different types of small towns should be clarified. The policy advantages of small towns with high development performance index values, such as Chengjiang Street, Harbor Development District, and the towns of Huashi and Xinqiao should be further strengthened to make them become the growth poles of small towns in Jiangyin. For small towns with general development performance index values such as Yunting Street and the town of Zhutang, system reforms and the strengthening of industrial supports should be actively implemented. For small towns with low development performance index values such as the towns of Changjing and Gushan, the policy compensation and support via funding and resources should be increased to accelerate a transformation through the mechanism of strong towns driving weak towns.

Second, considering the economic dimension, it is important that small towns with high development performance index values play a leading role in the gradual construction of an environmentally friendly industrial system guided by the concept of a circular economy. At present, some developed areas, such as Chengjiang Street, the High-Tech Zone, and the town of Xinqiao have implemented several initiatives to optimize and transform their industrial structure. However, in general, industry remains dominant in the economic development of small towns in Jiangyin. As the role of ecological factors in the development performance of small towns becomes increasingly prominent, it is necessary to grasp the degree of pollution emissions when introducing new industrial projects. Moreover, large-scale land expansion for economic construction should be stopped. Small towns with low development performance index values and optimal natural resource conditions can also develop ecological agriculture, leisure tourism, and other industries in combination, drawing on the "tourism +" development model implemented in the town of XuXiake to improve land use efficiency.

Finally, with regard to the social dimension, since many small towns other than Chengjiang Street do not have correspondingly perfect public transportation systems between each other, it is necessary to strengthen infrastructure development in small towns and improve the three levels of public transportation networks: urban, town, and inter-district bus systems. In addition, the social environment should also be taken into account, as careful design and manning of the townscape [94], for example, through wide streets and artificially shaped trees, can represent the modern town image. If small towns in economically developed areas such as Jiangyin can take the lead, they will serve as models for other regions and will have a significant impact on the development and transformation of small towns nationwide.

## 5. Conclusions

For this study, we selected 14 small towns in Jiangyin as the research units to study their development performance and influencing factors. The main conclusions of the study are presented as followed.

During the period from 2001–2019, the overall development performance of small towns in Jiangyin first decreased and then increased, with 2012 marking a turning point. From an individual perspective, the development performance showed three trends, which

are a stable overall increasing trend, an overall decreasing trend, and an initially decreasing and then increasing trend, respectively.

Our findings on the spatial evolution pattern from 2001 to 2019 revealed a fluctuating ascending process of complete dispersion → small agglomeration → large agglomeration, associated with the development performance index values of small towns in Jiangyin, which showed an optimized spatial pattern.

Lastly, GDP per capita, industrial investment, and construction land density were the main factors affecting the heterogeneous spatiotemporal evolution of small towns in Jiangyin. Economic and social factors had a strong driving effect on the development performance index values of small towns in Jiangyin and were the main factors influencing the spatial heterogeneity of these values. Ecological issues should receive constant attention in the future development of small towns.

**Author Contributions:** Conceptualization, methodology, formal analysis, investigation, writing—original draft preparation, X.G.; software, resources, data curation, visualization X.G. and J.T.; validation, supervision, project administration, X.G., H.L. and X.Z.; writing—review and editing, X.G., H.L., X.Z., J.T. and Y.Z.; funding acquisition, X.Z., H.L. and J.T. All authors have read and agreed to the published version of the manuscript.

**Funding:** This research was funded by National Natural Science Foundation of China, grant number: 42071224; National Social Science Foundation of China Post-grant Program, grant number: 21FSHB014; Humanities and Social Sciences Foundation of the Ministry of Education of China, grant number: 20YJCZH069 and 2022 Graduate Student Research and Practice Innovation Project of Jiangsu Province, China, grant number: 1812000024820.

**Institutional Review Board Statement:** Not applicable.

**Informed Consent Statement:** Not applicable.

**Data Availability Statement:** Not applicable.

**Conflicts of Interest:** The authors declare no conflict of interest.

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
