# Peer review of "An Evaluation of the Development Performance of Small County Towns and Its Influencing Factors: A Case Study of Small Towns in Jiangyin City in the Yangtze River Delta, China"

_land, doi:10.3390/land11071059_

Round 1

Reviewer 1 Report

The Abstract is relatively clear and concise. However, the authors should clarify (not only in the abstract) which is the study area. They first (from the title) talk about "small county towns"..."A Case Study of Jiangyin City", then they refer to Jiangyin as a region. Please clarify this. Somehow, this aspects are explained in sub-section 2.1, but should be clear from the title of the paper.
Lines 37-41 The authors should use scientific recommendations based on state-of-the-art literature instead of recommendations issued from political events.
The authors provide limited international scientific background in small towns evaluation. They just present urban performance evaluation (lines 58-65) briefly passing through a few international scientific works. A consistent focus is on the Chinese literature which, undoubtedly, is valuable, but not enough to present a scientific background of the issue at stake.

At the end of Introduction section, the authors could also clearly specify the objective(s) and research questions of the paper.
2.1. Should better clarify the type of urban structure of Jiangyin which seems to include 10 organic cities. Until this point, the reader doesn't quite understand if the focus is on Jiangyin town or the small towns which are part of it.
Materials and Methods.
The is no need to use a sub-sub-section 2.1.1.
First word of the line is incomplete.

Overall, the Methodology is clear and comprehensive.
Results and Analysis
3.1. You can just say "Evaluation of the Development Performance of Small Towns" to avoid repeatability
Table 2. In the "Subsystem" column you can just use Economic development and Social development since the System Layer is the Driving force. I think that is obvious that the two mentioned above are drivers. It goes the same with the Status. Just use Investment and construction. And the same with Response. Of course, this is just a suggestion, it is up to the authors to consider it.
Overall, Results and Analysis are quite comprehensive with lots of information and detailed analysis. It is recommended for the authors to double check the English and the way the causal explanations are provided to avoid repeatability and become more clear.
The Discussions are very interesting conceived, trying to address several aspects related to the reliability of the approach, the factors, policy recommendations. However,  the discussions should also consider similar experiences both in China and worldwide (if the case). They should discuss the current results in relation to what was written/found in other similar studies.

Conclusions are generally clear and succinct.

Reviewer 2 Report

It is a well-conceived and well-structured paper. The paper's subject concerning small towns is very interesting to promote its revitalization and urban-rural integration. However, this subject is maybe more suitable for the Urban Sciences Journal.

Only three remarks:

§  Subsections 3.1.3 and 3.2: considering 2012 as a turning point (subsections 3.1.1 and 3.1.2), the authors should be used a forward year (e.g. 2014 or 2015), to show the 2012 measures impacts;

§   Subsections 3.1.3 and 3.2: the authors should use 2019 instead of 2020, in order to dismiss the probable pandemic crisis effects;

§   Section 4.3: some recommendations are too generic and can be used anywhere (e.g. line 651 “…to strengthen inter-town collaboration and resource sharing.”; lines 654-655 “…further strengthened to attract population concentration, foreign investments, industrial transformation, and spatial expansion.”; lines 672-675 “…developing low-pollution green industries to reduce the burden on the ecological environment, and continuous innovation should be encouraged to develop green technologies for existing industrial enterprises along with strict control of pollution emissions.”; lines 680-683 “…to improve the social welfare system and the quality of public services and to retain the local population while attracting outsiders through the provision of good-quality education, employment, and medical security.”; line 683 “…to strengthen infrastructure development…”; lines 684-685 “…improve the three levels of public transportation networks: urban, town, and inter-district bus systems.”)

Minor revisions:

§   Lines 37-39: Please put the documents mentioned as references

§   Line 74: Please replace “…efficiency [40 41].” by “…efficiency [40–41].”

§   Line 76: Please replace “…projection method. and other…” by “…projection method, and other…”

§   Line 125: Please replace “iangyin is a riverside…” by “Jiangyin is a riverside…”

§   Figure 1: Please improve quality

§   Lines 154-155: Please put the document mentioned as reference

§   Table 1: Please put the Data Sources as references

§   Lines 189-192: Please replace “With xkm representing the input of the mth resource of the kth town, and yk n representing the output of the nth resource of the kth town (k = 1, 2, ...). ,K; m = 1, 2, ..., M; n = 1, 2, ..., N), the ith town…” by “With xkm representing the input of the mth resource of the kth town, and ykn representing the output of the nth resource of the kth town (k = 1, 2, ...). ,K; m = 1, 2, ..., M; n = 1, 2, ..., N), the ith town…”

§   Lines 213-217: Please replace “…X, n is the number of small towns in the study area, m is the number of types of influencing factors, ??,? is the number of small towns within type i for the influencing factor X, ?2? is the variance of the development performance index of small towns in the study area, and  is the variance of the development performance index of small towns in type i.” by “…X, n is the number of small towns in the study area, m is the number of types of influencing factors, ??,? is the number of small towns within type i for the influencing factor X, ?2? is the variance of the development performance index of small towns in the study area, and  is the variance of the development performance index of small towns in type i.”

§   Lines 270-272: To avoid misunderstandings, please explain better the sums of the weights because some indicators are negative

§   Lines 288-289: To avoid misunderstandings, please explain better the sums of the weights because some indicators are negative

§   Figure 2: Please use more contrasting colors

§   Line 361: Please replace “…Qingyang, XuXiake. Zhutang, and…” by “Qingyang, XuXiake, Zhutang, and…”

§   Lines 387-388: Please put the document as reference

§   Figure 4: Please improve quality

§   Lines 514-515: Please replace “…the 0.05 level for each year (Table 4).” by “…the 0.05 level for each year.”

§   Lines 527-528: Please replace “…with a q-value of 0.2532..” by “…with a q-value of 0.2328.”

§   Lines 530-531: Please rewrite the sentence because two ecological factors decrease but one of them increase
